# THE NATURAL LANGUAGE DECATHLON: MULTITASK LEARNING AS QUESTION ANSWERING

## ABSTRACT

Deep learning has improved performance on many natural language processing (NLP) tasks individually. However, general NLP models cannot emerge within a paradigm that focuses on the particularities of a single metric, dataset, and task. We introduce the Natural Language Decathlon (decaNLP), a challenge that spans ten tasks: question answering, machine translation, summarization, natural language inference, sentiment analysis, semantic role labeling, relation extraction, goal-oriented dialogue, semantic parsing, and commonsense pronoun resolution. We cast all tasks as question answering over a context. Furthermore, we present a new multitask question answering network (MQAN) that jointly learns all tasks in decaNLP without any task-specific modules or parameters more effectively than sequence-to-sequence and reading comprehension baselines. MQAN shows improvements in transfer learning for machine translation and named entity recognition, domain adaptation for sentiment analysis and natural language inference, and zero-shot capabilities for text classification. We demonstrate that the MQAN's multi-pointer-generator decoder is key to this success and that performance further improves with an anti-curriculum training strategy. Though designed for decaNLP, MQAN also achieves state of the art results on the WikiSQL semantic parsing task in the single-task setting. We also release code for procuring and processing data, training and evaluating models, and reproducing all experiments for decaNLP.

## 1 INTRODUCTION

We introduce the Natural Language Decathlon (decaNLP) in order to explore models that generalize to many different kinds of NLP tasks. decaNLP encourages a single model to simultaneously optimize for ten tasks: question answering, machine translation, document summarization, semantic parsing, sentiment analysis, natural language inference, semantic role labeling, relation extraction, goal oriented dialogue, and pronoun resolution.

We frame all tasks as question answering (Kumar et al., 2016) with a context, question, and answer (Fig. 1). Traditionally, NLP examples have inputs $x$ and outputs $y$, and the underlying task $t$ is provided through explicit modeling constraints. Meta-learning approaches include $t$ as additional input (Schmidhuber, 1987; Thrun and Pratt, 1998; Thrun, 1998; Vilalta and Drissi, 2002). Our approach does not use a single representation for any $t$, but instead uses the combination of natural language questions and contexts to orient the model to the correct task. This allows single models to effectively multitask, makes them more suitable as pretrained models, allows a model to generalize to completely new tasks through different but related contexts and questions.

We provide a set of baselines for decaNLP that combine the basics of sequence-to-sequence learning (Sutskever et al., 2014; Bahdanau et al., 2014; Luong et al., 2015b) with pointer networks (Vinyals et al., 2015; Merity et al., 2017; Gülçehre et al., 2016; Gu et al., 2016; Nallapati et al., 2016), advanced attention mechanisms (Xiong et al., 2017), attention networks (Vaswani et al., 2017), question answering (Seo et al., 2017; Xiong et al., 2018; Yu et al., 2016; Weissenborn et al., 2017), and curriculum learning (Bengio et al., 2009).

The multitask question answering network (MQAN) is designed for decaNLP and makes use of a novel dual coattention and multi-pointer-generator decoder to multitask across all tasks in decaNLP. Our results demonstrate that training the MQAN jointly on all tasks with the right anti-curriculum

**Examples**

| Question | Context | Answer | Question | Context | Answer |
|---|---|---|---|---|---|
| What is a major importance of Southern California in relation to California and the US? | ...Southern California is a major economic center for the state of California and the US.... | major economic center | What has something experienced? | Areas of the Baltic that have experienced eutrophication. | eutrophication |
| What is the translation from English to German? | Most of the planet is ocean water. | Der Großteil der Erde ist Meerwasser | Who is the illustrator of Cycle of the Werewolf? | Cycle of the Werewolf is a short novel by Stephen King, featuring illustrations by comic book artist Bernie Wrightson. | Bernie Wrightson |
| What is the summary? | Harry Potter star Daniel Radcliffe gains access to a reported £320 million fortune... | Harry Potter star Daniel Radcliffe gets £320M fortune... | What is the change in dialogue state? | Are there any Eritrean restaurants in town? | food: Eritrean |
| Hypothesis: Product and geography are what make cream skimming work. Entailment, neutral, or contradiction? | Premise: Conceptually cream skimming has two basic dimensions – product and geography. | Entailment | What is the translation from English to SQL? | The table has column names... Tell me what the notes are for South Australia | SELECT notes from table WHERE 'Current Slogan' = 'South Australia' |
| Is this sentence positive or negative? | A stirring, funny and finally transporting re-imagining of Beauty and the Beast and 1930s horror film. | positive | Who had given help? Susan or Joan? | Joan made sure to thank Susan for all the help she had given. | Susan |

Figure 1: Overview of the decaNLP dataset with one example from each decaNLP task in the order presented in Section 2. Each task is framed as a form of question answering. Answer words in red are generated by pointing to the context, in green from the question, and in blue if they are generated from a classifier over the full output vocabulary.

strategy can achieve performance comparable to that of ten separate MQANs, each trained separately. A MQAN pretrained on decaNLP shows improvements in transfer learning for machine translation and named entity recognition, domain adaptation for sentiment analysis and natural language inference, and zero-shot capabilities for text classification. Though not explicitly designed for any one task, MQAN proves to be a strong model in the single-task setting as well, achieving state-of-the-art results on the semantic parsing component of decaNLP.

We have released all code[1] used for this project as well as a leaderboard[2] based on decathlon scores (decaScore). We hope that the combination of these resources will facilitate research in multitask learning, transfer learning, general embeddings and encoders, architecture search, zero-shot learning, general purpose question answering, meta-learning, and other related areas of NLP.

## 2 TASKS AND METRICS

decaNLP consists of 10 publicly available datasets with examples cast as (question, context, answer) triplets as shown in Fig. 1. For a detailed discussion of why these ten tasks were chosen over others, please refer to Appendix A.

**Question Answering.** Question answering (QA) models receive a question and a context that contains information necessary to output the desired answer. We use the Stanford Question Answering Dataset (SQuAD) (Rajpurkar et al., 2016) for this task. Contexts are paragraphs taken from the English Wikipedia, and answers are sequences of words copied from the context. SQuAD uses a normalized F1 (nF1) metric that strips out articles and punctuation.

**Machine Translation.** Machine translation models receive an input document in a source language that must be translated into a target language. We use the 2016 English to German training data prepared for the International Workshop on Spoken Language Translation (IWSLT) (Cettolo et al., 2016). We evaluate with a corpus-level BLEU score (Papineni et al., 2002) on the 2013 and 2014 test sets as validation and test sets, respectively.

**Summarization.** Summarization models take in a document and output a summary of that document. We used the transformed, non-anonymized version of the CNN/DailyMail (CNN/DM) corpus (Hermann et al., 2015) by dataset (Nallapati et al., 2016). We average ROUGE-1, ROUGE-2, and ROUGE-L scores (Lin, 2004) to compute an overall ROUGE score.

**Natural Language Inference.** Natural Language Inference (NLI) models receive two input sentences: a premise and a hypothesis. Models must then classify the inference relationship between the two as one of entailment, neutrality, or contradiction. We use the Multi-Genre Natural Language Inference Corpus (MNLI) (Williams et al., 2017) which provides training examples from multiple

---

[1] anonymizedurl.com

[2] anonymizedurl.com

Table 1: Summary of openly available benchmark datasets in decaNLP and evaluation metrics that contribute to the decaScore. All metrics are case insensitive. nF1 is a normalized F1 metric that strips out articles and punctuation. EM is an exact match comparison: for text classification, this amounts to accuracy; for WOZ it is equivalent to turn-based dialogue state exact match (dsEM) and for WikiSQL it is equivalent to exact match of logical forms (lfEM). F1 for QA-ZRE is a corpus level metric (cF1) that takes into account that some questions are unanswerable.

| Task | Dataset | # Train | # Dev | # Test | Metric |
|------|---------|---------|-------|--------|--------|
| Question Answering | SQuAD | 87599 | 10570 | 9616 | nF1 |
| Machine Translation | IWSLT | 196884 | 993 | 1305 | BLEU |
| Summarization | CNN/DM | 287227 | 13368 | 11490 | ROUGE |
| Natural Language Inference | MNLI | 392702 | 20000 | 20000 | EM |
| Sentiment Analysis | SST | 6920 | 872 | 1821 | EM |
| Semantic Role Labeling | QA-SRL | 6414 | 2183 | 2201 | nF1 |
| Zero-Shot Relation Extraction | QA-ZRE | 840000 | 600 | 12000 | cF1 |
| Goal-Oriented Dialogue | WOZ | 2536 | 830 | 1646 | dsEM |
| Semantic Parsing | WikiSQL | 56355 | 8421 | 15878 | lfEM |
| Pronoun Resolution | MWSC | 80 | 82 | 100 | EM |

domains (transcribed speech, popular fiction, government reports) and test pairs from seen and unseen domains. MNLI uses an exact match (EM) score.

**Sentiment Analysis.** Sentiment analysis models are trained to classify the sentiment expressed by input text. The Stanford Sentiment Treebank (SST) (Socher et al., 2013) consists of movie reviews with the corresponding sentiment (positive, neutral, negative). We use the unparsed, binary version (Radford et al., 2017). SST also uses an EM score.

**Semantic Role Labeling.** Semantic role labeling (SRL) models are given a sentence and predicate (typically a verb) and must determine 'who did what to whom,' 'when,' and 'where' (Johansson and Nugues, 2008). We use an SRL dataset that treats the task as question answering, QA-SRL (He et al., 2015). This dataset covers both news and Wikipedia domains, but we only use the latter in order to ensure that all data for decaNLP can be freely downloaded. We evaluate QA-SRL with the nF1 metric used for SQuAD.

**Relation Extraction.** Relation extraction systems take in a piece of unstructured text and the kind of relation that is to be extracted from that text. As with SRL, we use a dataset that maps relations to a set of questions so that relation extraction can be treated as question answering: QA-ZRE (Levy et al., 2017). Evaluation of the dataset is designed to measure zero shot performance on new kinds of relations – the dataset is split so that relations seen at test time are unseen at train time. This kind of zero-shot relation extraction, framed as question answering, makes it possible to generalize to new relations. QA-ZRE uses a corpus-level F1 metric (cF1) in order to accurately account for when relations are not present, in which case the question is unanswerable.

**Goal-Oriented Dialogue.** Dialogue state tracking is a key component of goal-oriented dialogue systems. Based on user utterances and actions taken, dialogue state trackers keep track of which user goals and requests as the system and user interact turn-by-turn. We use the English Wizard of Oz (WOZ) restaurant reservation task (Wen et al., 2016), which comes with a predefined ontology of foods, dates, times, addresses, and other information that would help an agent make a reservation for a customer. WOZ is evaluated by turn-based dialogue state EM (dsEM) over the goals of the customers.

**Semantic Parsing.** SQL query generation is related to semantic parsing. Models based on the WikiSQL dataset (Zhong et al., 2017) translate natural language questions into structured SQL queries so that users can interact with a database in natural language. WikiSQL is evaluated by a logical form exact match (lfEM) to ensure that models do not obtain correct answers from incorrectly generated queries.

**Pronoun Resolution.** Our final task is based on Winograd schemas (Winograd, 1972), which require pronoun resolution: "Joan made sure to thank Susan for the help she had [given/received]. Who had [given/received] help? Susan or Joan?". We started with examples taken from the Winograd Schema Challenge (Levesque et al., 2011) and modified them to ensure that answers were a single word from

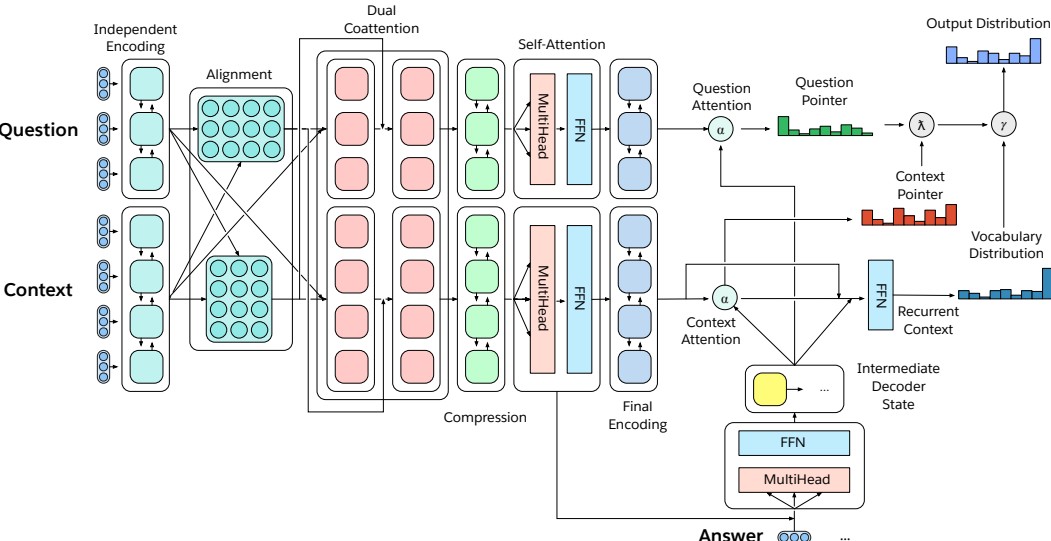

Figure 2: Overview of the MQAN model. It takes in a question and context document, encodes both with a BiLSTM, uses dual coattention to condition representations for both sequences on the other, compresses all of this information with another two BiLSTMs, applies self-attention to collect long-distance dependency, and then uses a final two BiLSTMs to get representations of the question and context. The multi-pointer-generator decoder uses attention over the question, context, and previously output tokens to decide whether to copy from the question, copy from the context, or generate from a limited vocabulary.

the context. This modified Winograd Schema Challenge (MWSC) ensures that scores are neither inflated nor deflated by oddities in phrasing or inconsistencies between context, question, and answer. We evaluate with an EM score.

**The Decathlon Score (decaScore).** Models competing on decaNLP are evaluated using an additive combination of each task-specific metric. All metrics fall between $0$ and $100$, so that the decaScore naturally falls between $0$ and $1000$ for ten tasks. Using an additive combination avoids issues that arise from weighing different metrics. All metrics are case insensitive.

## 3 MULTITASK QUESTION ANSWERING NETWORK (MQAN)

Because every task is framed as question answering and trained jointly, we call our model a multitask question answering network (MQAN). Each example consists of a context, question, and answer as shown in Fig. 1. Many recent QA models for question answering typically assume the answer can be copied from the context (Wang and Jiang, 2017; Seo et al., 2017; Xiong et al., 2018), but this assumption does not hold for general question answering. The question often contains key information that constrains the answer space. Noting this, we extend the coattention of (Xiong et al., 2017) to enrich the representation of not only the input but also the question. Also, the pointer-mechanism of (See et al., 2017) is generalized into a hierarchical, multi-pointer-generator that enables the capacity to copy directly from the question and the context.

During training, the MQAN takes as input three sequences: a context $c$ with $l$ tokens, a question $q$ with $m$ tokens, and an answer $a$ with $n$ tokens. Each of these is represented by a matrix where the $i$th row of the matrix corresponds to a $d_{emb}$-dimensional embedding (such as word or character vectors) for the $i$th token in the sequence:

$$C \in \mathbb{R}^{l \times d_{emb}} \qquad Q \in \mathbb{R}^{m \times d_{emb}} \qquad A \in \mathbb{R}^{n \times d_{emb}} \tag{1}$$

An encoder takes these matrices as input and uses a deep stack of recurrent, coattentive, and self-attentive layers to produce final representations, $C_{fin} \in \mathbb{R}^{l \times d}$ and $Q_{fin} \in \mathbb{R}^{m \times d}$, of both context and question sequences designed to capture local and global interdependencies. Appendix E describes the full details of the encoder.

**Answer Representations.** During training, the decoder begins by projecting the answer embeddings onto a $d$-dimensional space:

$$AW_2 = A_{proj} \in \mathbb{R}^{n \times d} \tag{2}$$

This is followed by a self-attentive layers, which has a corresponding self-attentive layer in the encoder. Because it lacks both recurrence and convolution, we add to $A_{proj}$ positional encodings (Vaswani et al., 2017) $PE \in \mathbb{R}^{n \times d}$ with entries

$$PE[t, k] = \begin{cases} \sin(t/10000^{k/2d}) & k \text{ is even} \\ \cos(t/10000^{(k-1)/2d}) & k \text{ is odd} \end{cases} \qquad A_{proj} + PE = A_{ppr} \in \mathbb{R}^{n \times d} \tag{3}$$

**Multi-head Decoder Attention.** We use self-attention[3] (Vaswani et al., 2017) so that the decoder is aware of previous outputs (or a special intialization token in the case of no previous outputs) and attention over the context to prepare for the next output. Refer to Appendix E for definitions of MultiHead attention and FFN, the residual feedforward network applied after MultiHead attention over the context.

$$\text{MultiHead}_A(A_{ppr}, A_{ppr}, A_{ppr}) = A_{mha} \in \mathbb{R}^{n \times d} \tag{4}$$

$$\text{MultiHead}_{AC}((A_{mha} + A_{ppr}), C_{fin}, C_{fin}) = A_{ac} \in \mathbb{R}^{n \times d} \tag{5}$$

$$FFN_A(A_{ac} + A_{mha} + A_{ppr}) = A_{self} \in \mathbb{R}^{n \times d} \tag{6}$$

**Intermediate Decoder State.** We next use a standard LSTM with attention to get a recurrent context state $\tilde{c}_t$ for time-step $t$. First, the LSTM produces an intermediate state $h_t$ using the previous answer word $A_{self}^{t-1}$ and recurrent context state (Luong et al., 2015b):

$$\text{LSTM}([(A_{self})_{t-1} ; \tilde{c}_{t-1}], h_{t-1}) = h_t \in \mathbb{R}^d \tag{7}$$

**Context and Question Attention.** This intermediate state is used to get attention weights $\alpha_t^C$ and $\alpha_t^Q$ to allow the decoder to focus on encoded information relevant to time step $t$.

$$\text{softmax}C_{fin}(W_2 h_t) = \alpha_t^C \in \mathbb{R}^l \qquad \text{softmax}Q_{fin}(W_3 h_t) = \alpha_t^Q \in \mathbb{R}^m \tag{8}$$

**Recurrent Context State.** Context representations are combined with these weights and fed through a feedforward network with tanh activation to form the recurrent context state and question state:

$$\tanh\left(W_4 \left[C_{fin}^\top \alpha_t^C ; h_t\right]\right) = \tilde{c}_t \in \mathbb{R}^d \qquad \tanh\left(W_5 \left[Q_{fin}^\top \alpha_t^Q ; h_t\right]\right) = \tilde{q}_t \in \mathbb{R}^d \tag{9}$$

**Multi-Pointer-Generator.** Our model must be able to generate tokens that are not in the context or the question. We give it access to $v$ additional vocabulary tokens. We obtain distributions over tokens in the context, question, and this external vocabulary, respectively, as

$$\sum_{i:c_i = w_t} \left(\alpha_t^C\right)_i = p_c(w_t) \in \mathbb{R}^n \qquad \sum_{i:q_i = w_t} \left(\alpha_t^Q\right)_i = p_q(w_t) \in \mathbb{R}^m \tag{10}$$

$$\text{softmax}W_v \tilde{c}_t = p_v(w_t) \in \mathbb{R}^v \tag{11}$$

These distributions are extended to cover the union of the tokens in the context, question, and external vocabulary by setting missing entries in each to 0 so that each distribution is in $\mathbb{R}^{l+m+v}$. Two scalar switches regulate the importance of each distribution in determining the final output distribution.

$$\sigma\left(W_{pv}\left[\tilde{c}_t ; h_t ; (A_{self})_{t-1}\right]\right) = \gamma \in [0, 1] \qquad \sigma\left(W_{cq}\left[\tilde{q}_t ; h_t ; (A_{self})_{t-1}\right]\right) = \lambda \in [0, 1] \tag{12}$$

$$\gamma p_v(w_t) + (1 - \gamma)\left[\lambda p_c(w_t) + (1 - \lambda)p_q(w_t)\right] = p(w_t) \in \mathbb{R}^{l+m+v} \tag{13}$$

We train using a token-level negative log-likelihood loss over all time-steps: $\mathcal{L} = -\sum_t^T \log p(a_t)$.

Table 2: Validation metrics for decaNLP baselines: sequence-to-sequence (S2S) with self-attentive transformer layers (+SAtt), the addition of coattention (+CAtt) over a split context and question, and a question pointer (+QPtr). The last model is equivalent to MQAN. Multitask models use a round-robin batch-level sampling strategy to jointly train on the full decaNLP. The last column includes an additional anti-curriculum (+ACurr) phase that trains on SQuAD alone before switching to the fully joint strategy. Entries marked with '-' would correspond to decaScores for aggregates of separately trained models; this is not well-defined without a mechanism for choosing between models.

| Dataset | Single-task Training | | | | Multitask Training | | | | |
| | S2S | +SAtt | +CAtt | +QPtr | S2S | +SAtt | +CAtt | +QPtr | +ACurr |
|---|---|---|---|---|---|---|---|---|---|
| SQuAD | 48.2 | 68.2 | 74.6 | 75.3 | 47.5 | 66.8 | 71.8 | 70.8 | 74.4 |
| IWSLT | 25.0 | 23.3 | 26.0 | 26.7 | 14.2 | 13.6 | 9.0 | 16.1 | 18.6 |
| CNN/DM | 19.0 | 20.0 | 25.1 | 25.5 | 25.7 | 14.0 | 15.7 | 23.9 | 24.3 |
| MNLI | 67.5 | 68.5 | 34.7 | 73.0 | 60.9 | 69.0 | 70.4 | 70.5 | 71.5 |
| SST | 86.4 | 86.8 | 86.2 | 88.5 | 85.9 | 84.7 | 86.5 | 86.2 | 87.4 |
| QA-SRL | 63.5 | 67.8 | 74.8 | 77.9 | 68.7 | 75.1 | 76.1 | 75.8 | 78.4 |
| QA-ZRE | 20.0 | 19.9 | 16.6 | 24.3 | 28.5 | 31.7 | 28.5 | 28.0 | 37.6 |
| WOZ | 85.3 | 86.0 | 86.5 | 88.0 | 84.0 | 82.8 | 75.1 | 80.6 | 84.8 |
| WikiSQL | 60.0 | 72.4 | 72.3 | 73.5 | 45.8 | 64.8 | 62.9 | 62.0 | 64.8 |
| MWSC | 43.9 | 46.3 | 40.4 | 48.8 | 52.4 | 43.9 | 37.8 | 48.8 | 48.8 |
| decaScore | - | - | - | - | 513.6 | 546.4 | 533.8 | 562.7 | **590.6** |

## 4 EXPERIMENTS AND ANALYSIS

### 4.1 BASELINES AND MQAN

In our framework, training examples are (question, context, answer) triplets. Our first baseline is the pointer-generator sequence-to-sequence (S2S) model of See et al. (2017), modified only to take in fixed GloVe vectors instead of training word vectors from scratch. S2S models take in only a single input sequence, so we concatenate the context and question for this model. In Table 2, validation metrics reveal that the S2S model does not perform well on SQuAD. On WikiSQL, it obtains a much higher score than prior sequence-to-sequence baselines (Zhong et al., 2017), but it is low compared to MQAN (+QPtr) and other baselines.

Augmenting the S2S model with self-attentive (+SAtt) encoder and decoder layers Vaswani et al. (2017), as detailed in E, increases the model's capacity to integrate information from both context and question. This improves performance on SQuAD by 20 nF1, QA-SRL by 4 nF1, and WikiSQL by 12 LFEM. For WikiSQL, this model nearly matches the prior state-of-the-art validation results of 72.4% without using a structured approach (Dong and Lapata, 2018; Huang et al., 2018; Yu et al., 2018b).

We next explore splitting the context and question into two input sequences as in typical reading comprehension and question answering settings. We augment the S2S model with a coattention mechanism (+CAtt) from reading comprehension models to tackle this new task formulation. Performance on SQuAD and QA-SRL increases by more than 5 nF1 each. Unfortunately, this fails to improve other tasks, and it significantly hurts performance on MNLI and MWSC. For these two tasks, answers can be copied directly from the question. Because both S2S baselines had the question concatenated to the context, the pointer-generator mechanism was able to copy directly from the question. When the context and question were separated into two different inputs, the model lost this ability.

To remedy this, we add a question pointer (+QPtr) to the previous baseline, which gives the MQAN described in Section 3 and Appendix E. This boosts performance on both MNLI and MWSC above prior baselines. It also improved performance on SQuAD to 75.5 nF1, which matches performance of the first wave of SQuAD models to make use of direct span supervision (Xiong et al., 2017). This

---

[3]The decoder operates step by step. To prevent the decoder from seeing future time-steps during training, appropriate entries of $XY^\top$ are set to a large negative number prior to the softmax in equation 22.

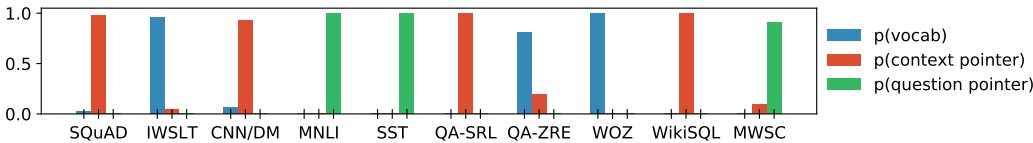

Figure 3: An analysis of how the MQAN chooses to output answer words. When p(generation) is highest, the MQAN places the most weight on the external vocab. When p(context) is highest, the MQAN places the most weight on the pointer distribution over the context. When p(question) is highest, the MQAN places the most weight on the pointer distribution over the question.

makes it the highest performing question answering model trained on SQuAD that does not explicitly model the problem as span extraction.

This last model achieved a new state-of-the-art test result on WikiSQL by reaching 72.4% lfEM and 80.4% database execution accuracy, surpassing the previous state of the art set by (Dong and Lapata, 2018) at 71.7% and 78.5%.

In the multitask setting, we see similar results, but we also notice several additional striking features. QA-ZRE performance increases 11 F1 points over the highest single-task models, which supports the hypothesis that multitask learning can lead to better generalization for zero-shot learning. See Appendix D for details regarding pre-processing and hyperparameters. See Appendix G for a deeper analysis of how different tasks are related and contribute to the decaScore as well as further experiments using contextualized word vectors (McCann et al., 2017).

### 4.2 OPTIMIZATION STRATEGIES AND CURRICULUM LEARNING

For multitask training, we experiment with various round-robin batch-level sampling strategies. Fully joint training cycles through all tasks from the beginning of training. However, some tasks require more iterations to converge in the single-task setting, which suggests that these are more difficult for the model to learn. We experiment with both curriculum and anti-curriculum strategies Bengio et al. (2009) based on this notion of difficulty.

We divide tasks into two groups: the easiest difficult task requires more than twice the iterations the most difficult easy task requires. Compared to the fully joint strategy, curriculum learning jointly trains the easier tasks (SST, QA-SRL, QA-ZRE, WOZ, WikiSQL, and MWSC) first. This leads to a dramatically reduced decaScore (Appendix F). Anti-curriculum strategies boost performance on tasks trained early, but can also hurt performance on tasks held out until later training. Of the various anti-curriculum strategies we experimented with, only the one which trains on SQuAD alone before transitioning to a fully joint strategy yielded a decaScore higher than using the fully joint strategy without modification. For a full comparison, see Appendix F.

### 4.3 ANALYSIS

**Multi-Pointer-Generator and task identification.** At each step, the MQAN decides between three choices: generating from the vocabulary, pointing to the question, and pointing to the context. While the model is not trained with *explicit* supervision for these decisions, it learns to switch between the three options. Fig. 3 presents statistics of how often the final model chooses each option. For SQuAD, QA-SRL, and WikiSQL, the model mostly copies from the context. This is intuitive because all tokens necessary to correctly answer questions from these datasets are contained in the context. The model also usually copies from the context for CNN/DM because answer summaries consist mostly of words from the context with few words generated from outside the context in between.

For SST, MNLI, and MWSC, the model prefers the question pointer because the question contains the tokens for acceptable classes. Because the model learns to use the question pointer in this way, it can do zero-shot classification as discussed in 4.3. For IWSLT and WOZ, the model prefers generating from the vocabulary because German words and dialogue state fields are rarely in the context. The models also avoids copying for QA-ZRE; half of those examples require generating 'unanswerable' from the external vocabulary.

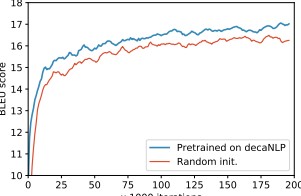 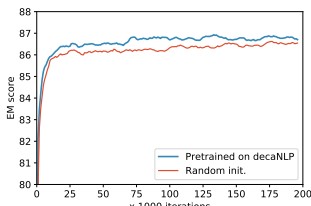

Figure 4: MQAN pretrained on decaNLP outperforms random initialization when adapting to new domains and learning new tasks. Left: training on a new language pair – English to Czech, right: training on a new task – Named Entity Recognition (NER).

Sampled answers confirm that the model does not confuse tasks. German words are only ever output during translation from English to German. The model never outputs anything but 'positive' and 'negative' for sentiment analysis.

**Adaptation to new tasks.** MQAN trained on decaNLP learn to generalize beyond the specific domains for any one task while also learning representations that make learning completely new tasks easier. For two new tasks (English-to-Czech translation and named entity recognition - NER), fine-tuning a MQAN trained on decaNLP requires fewer iterations and reaches a better final performance than training from a random initialization (Fig. 4). For the translation experiment, we use the IWSLT 2016 En→Cs dataset and for NER, we use OntoNotes 5.0 (Hovy et al., 2006). For both of these experiments, we retain the model weights and only train a (new) softmax layer that contains the necessary tokens for the new tasks.

**Zero-shot domain adaptation for text classification.** Because MNLI is included in decaNLP, it is possible to adapt to the related Stanford Natural Language Inference Corpus (SNLI) (Bowman et al., 2015) without changing the model at all. Fine-tuning a MQAN pretrained on decaNLP and training exactly as before on MultiNLI achieves an 87% test exact match score, which is a 2% increase over training from a random initialization and 2% from the state of the art (Kim et al., 2018). Remarkably, without any training on SNLI, a MQAN pretrained on decaNLP still achieves an EM score of 62%. Because decaNLP contains SST, it can also perform well on other binary sentiment classification tasks without any changes to the model or fine-tuning. We used Amazon and Yelp reviews (Kotzias et al., 2015) as an out of domain test set. A MQAN pretrained on decaNLP achieves test exact match scores of 82.1% and 80.8%, respectively, without any fine-tuning.

Additionally, rephrasing questions by replacing the tokens for the training labels *positive/negative* with *happy/angry* or *supportive/unsupportive* at inference time, leads to only small degradation in performance. The model's reliance on the question pointer for SST (see Figure 3) allows it to copy different, but related class labels with little confusion. This suggests these multitask models are more robust to slight variations in questions and tasks and can generalize to new and unseen classes.

These results demonstrate that models trained on decaNLP have the potential to simultaneously generalize to out-of-domain contexts and questions for multiple tasks and adapt to unseen classes for text classification. This zero-shot domain input and output spaces suggests that the breadth of tasks in decaNLP encourages generalization beyond what can be achieved by training for a single task.

## 5 CONCLUSION

We introduced the Natural Language Decathlon (decaNLP), a new benchmark for measuring the performance of NLP models across ten tasks that appear disparate until unified as question answering. We presented MQAN, a model for general question answering that uses a multi-pointer-generator decoder to capitalize on questions as natural language descriptions of tasks. Despite not having any task-specific modules, we trained MQAN on all decaNLP tasks jointly, and we showed that anti-curriculum learning gave further improvements. After training on decaNLP , MQAN exhibits transfer learning and zero-shot capabilities. When used as pretrained weights, MQAN improved performance on new tasks. It also demonstrated zero-shot domain adaptation capabilities on text classification from new domains. We hope the the decaNLP benchmark, experimental results, and publicly available code encourage further research into general models for NLP.

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

# A   TASK MOTIVATIONS

The Natural Language Decathlon asks whether we have learned enough from single tasks to get a sense of how much of natural language current methods really understand. With this in mind, we have several intentions for models that attempt the Decathlon, and we have chosen the tasks in such a way that they reflect these intentions. Models should be able to:

1. interact with people regardless of their natural language,
2. work well across many different domains of natural language,
3. extract information about mental states from natural language,
4. summarize what is understood,
5. answer questions about specific pieces of text and retrieve pertinent information,
6. convey when they have insufficient information to answer questions,
7. understand semantic relationships related to the roles and actions in the world,
8. interact with other machines,
9. perform linguistic-based reasoning that is easy for humans,
10. interact with humans to achieve a goal,
11. convey relevant information in a human readable format,
12. learn relatedness of tasks to allow for zero-shot adjustment to new tasks

Noticeably, we do not include an intention for models to understand linguistic features explicitly. There are two reasons for this. First, humans demonstrate that it is possible to satisfy all of the above intentions without an explicit linguistic understanding of natural language. Second, it is already understood how tasks like part-of-speech tagging, parsing, chunking, etc. can contribute to models performing higher-level tasks (Hashimoto et al., 2016). For the latter reason, we highly encourage experimentation with intermediate tasks that might aid models in decaNLP.

The final intention deals more strongly with the specific approach to decaNLP used in this paper than it does with decaNLP itself. This is in line with our belief that we need to move away from hand-designed parameter sharing and transfer learning. In the same way that moving away from hand-crafted features to learned features made new things possible, we believe that we should let the model decide how to distribute its knowledge. This is in an effort to ensure that we are not limiting the model's ability to generalize to new tasks by cutting off helpful signal from any previously learned tasks.

## B   RELATED WORK

This section contains work related to aspects of decaNLP and MQAN that are not task-specific. See Appendix C for work related to each individual task.

**Transfer Learning in NLP.**   Most success in making use of the relatedness between natural language tasks stem from transfer learning. Word2Vec (Mikolov et al., 2013a;b), skip-thought vectors (Kiros et al., 2015) and GloVe (Pennington et al., 2014) yield pretrained embeddings that capture useful information about natural language. The embeddings (Collobert and Weston, 2008; Collobert et al., 2011), intermediate representations (Peters et al., 2018), and weights of language models can be transferred to similar architectures (Ramachandran et al., 2017) and classification tasks (Howard and Ruder, 2018). Intermediate representations from supervised machine translation models improve performance on question answering, sentiment analysis, and natural language inference (McCann et al., 2017). Question answering datasets support each other as well as entailment tasks (Min et al., 2017), and high-resource machine translation can support low-resource machine translation (Zoph et al., 2016). This work shows that the combination of MQAN and decaNLP makes it possible to transfer an entire end-to-end model that can be adapted for any NLP task cast as question answering.

**Multitask Learning in NLP.**   Unified architectures have arisen for chunking, POS tagging, NER, and SRL (Collobert et al., 2011) as well as dependency parsing, semantic relatedness, and natural language inference (Hashimoto et al., 2016). Multitask learning over different machine translation language pairs can enable zero-shot translation (Johnson et al., 2017), and sequence-to-sequence architectures can be used to multitask across translation, parsing, and image captioning (Luong et al., 2015a) using varying numbers of encoders and decoders. These tasks can also be learned with image classification and speech recognition with careful modularization (Kaiser et al., 2017), and the success of this approach extends to visual and textual question answering (Xiong et al., 2016). Learning such modularization can further mitigate interference between tasks (Ruder et al., 2017).

More generally, multitask learning has been successful when models are able to capitalize on relatedness amongst tasks while mitigating interference from dissimilarities (Caruana, 1997). When tasks are sufficiently related, they can provide an inductive bias (Mitchell, 1980) that forces models to learn more generally useful representations. By unifying tasks under a single perspective, it is possible to explore these relationships (Wang et al., 2018; Poliak et al., 2018a;b).

MQAN trained on decaNLP is the first, single model to achieve reasonable performance on such a wide variety of complex NLP tasks without task-specific modules or parameters, with little evidence of catastrophic interference, and without parse trees, chunks, POS tags, or other intermediate representations. This sets the foundation for general question answering models.

**Optimization and Catastrophic Forgetting.**   Multitask learning presents a set of optimization problems that extend beyond the NLP setting. Multi-objective optimization (Deb, 2014) naturally connects to multitask learning and typically involves querying a decision-maker who weighs different objectives. Much effort has gone into mitigating catastrophic forgetting (McCloskey and Cohen, 1989; Ratcliff, 1990; Kemker et al., 2017) by penalizing the norm of parameters when training on a new task (Kirkpatrick et al., 2017), the norm of the difference between parameters for previously learned tasks during parameter updates (Hashimoto et al., 2016), incrementally matching modes (Lee et al., 2017), rehearsing on old tasks (Robins, 1995), using adaptive memory buffers (Gepperth and Karaoguz, 2016), finding task-specific paths through networks (Fernando et al., 2017), and packing new tasks into already trained networks (Mallya and Lazebnik, 2017).

MQAN is able to perform nearly as well or better in the multitask setting as in the single-task setting for each task despite being capped at the same number of trainable parameters in both. A collection of MQANs trained for each task individually would use far more trainable parameters than a single MQAN trained jointly on decaNLP. This suggests that MQAN successfully uses trainable parameters more efficiently in the multitask setting by learning to pack or share parameters in a way that limits catastrophic forgetting.

**Meta-Learning**   Meta-learning attempts to train models on a variety of tasks so that they can easily learn new tasks (Thrun and Pratt, 1998; Thrun, 1998; Vilalta and Drissi, 2002). Past work has shown how to learn rules for learning (Schmidhuber, 1987; Bengio et al., 1992), train meta-agents that

control parameter updates (Hochreiter et al., 2001; Andrychowicz et al., 2016), augment models with special memory mechanisms (Santoro et al., 2016; Schmidhuber, 1992), and maximize the degree to which models can learn new tasks (Finn et al., 2017).

## C Task-Specific Related Work

**Question Answering.** Early success on the SQuAD dataset exploited the fact that all answers can be found verbatim in the context. State-of-the-art models point to start and end tokens in the document (Seo et al., 2017; Xiong et al., 2017; Yu et al., 2016; Weissenborn et al., 2017). This allowed deterministic answer extraction to overtake sequential token generation (Wang and Jiang, 2017). This quirk of the dataset does not hold for question answering in general, so recent models for SQuAD are not necessarily general question answering models (Yu et al., 2018a; Hu et al., 2018; Wang et al., 2017a; Liu et al., 2017b; Huang et al., 2017; Xiong et al., 2018; Liu et al., 2017a; Pan et al., 2017; Salant and Berant, 2017). While datasets like TriviaQA (Joshi et al., 2017) and NewsQA (Trischler et al., 2017) could also represent question answering, SQuAD is particularly interesting because the human level performance of SQuAD models in the single-task setting depends on a quirk that does not generalize to all forms of question answering. Including SQuAD in decaNLP challenges models to integrate techniques learned from a single-task approach into a more general approach while evaluation remains grounded in the document. Many of the alternatives are larger and can be used as additional training data or incorporated into future iterations of the decaNLP once the more well-understood SQuAD dataset has been mastered in the multitask setting.

**Machine Translation.** Until recently, the standard approach trained recurrent models with attention (Luong et al., 2015b; Bahdanau et al., 2014) on a single source-target language pair (Wu et al., 2016; Sennrich et al., 2017). Models that use only convolution (Gehring et al., 2017) or attention (Vaswani et al., 2017) have shown that recurrence is not essential for the task, but recurrence can contribute to the strongest models (Chen et al., 2018). While training these models on many source and target languages at the same time remains difficult, limiting models to one source language and many target languages or vice versa can lead to strong performance when resources are limited or null (Johnson et al., 2017).

While much larger corpora and many other language pairs exist, the English-German IWSLT dataset provides the same order of magnitude of training data as the other tasks in decaNLP. We encourage the use of larger corpora or multiple language pairs to improve performance, but we did not want to skew the first iteration of the challenge too far towards machine translation.

**Summarization** Recent approaches combine recurrent neural networks with pointer networks to generate output sequences that contain key words copied from the document (Nallapati et al., 2016). Coverage mechanisms (Nallapati et al., 2016; See et al., 2017; Suzuki and Nagata, 2017) and temporal attention (Paulus et al., 2017) improve problems with redundancy in long summaries. Reinforcement learning has pushed performance using common summarization metrics (Paulus et al., 2017) as well as alternative metrics that transfer knowledge from another task (Pasunuru et al., 2017; Pasunuru and Bansal, 2018).

While new corpora like NEWSROOM (Grusky et al., 2018) are even larger, CNN/DM remains the current standard benchmark, so we include it in decaNLP and encourage augmentation with datasets like NEWSROOM.

**Natural Language Inference** NLI has a long history playing roles in tasks like information retrieval and semantic parsing (Fyodorov et al., 2000; Condoravdi et al., 2003; Bos and Markert, 2005; Dagan et al., 2005; MacCartney and Manning, 2009). The introduction of the Stanford Natural Language Inference Corpus (SNLI) by (Bowman et al., 2015) spurred a new wave of interest in NLI, its connections to other tasks, and general sentence representations. The most successful approaches make use of attentional models that match and align words in the premise to those in the hypothesis (Tay et al., 2017; Peters et al., 2018; Ghaeini et al., 2018; Chen et al., 2017; Wang et al., 2017b; McCann et al., 2017), but recent non-attentional models designed to extract useful sentence representations have nearly closed the gap (Liu et al., 2017b; Im and Cho, 2017; Shen et al., 2018; Choi et al., 2017).

The dataset we use, the Multi-Genre Natural Language Inference Corpus (MNLI) introduced by (Williams et al., 2017), is the successor to SNLI. Recent approaches to MNLI use methods developed on SNLI and have even pointed out the similarities between models for question answering and NLI (Huang et al., 2017).

**Sentiment Analysis**    Because SST came with parse trees for every example, some approaches use all of the sub-tree labels by modeling trees explicitly (Yu and Munkhdalai, 2017b; Tai et al., 2015) as in the original paper. Others use sub-tree labels implicitly (Yu and Munkhdalai, 2017a; McCann et al., 2017; Peters et al., 2018), and still others do not use the sub-trees at all (Radford et al., 2017). This suggests that while the many sub-tree labels might facilitate learning, they are not necessary to train state-of-the-art models.

**Semantic Role Labeling**    Traditionally, models have made use of syntactic parsing information Punyakanok et al. (2008), but recent methods have demonstrated that it is not necessary to use syntactic information as additional input (Zhou and Xu, 2015; Marcheggiani et al., 2017). State-of-the-art approaches treat SRL as a tagging problem (He et al., 2017), make use of that specific structure to constrain decoding, and mix recurrent and self-attentive layers (Tan et al., 2017).

Because QA-SRL treats SRL as question answering (He et al., 2015), it abstracts away the many task-specific constraints of treating SRL as a tagging problem with hand-designed verb-specific roles or grammars. This preserves much of the structure extracted by prior formulations while also allowing models to extract structure that is not syntax-based.

**Relation Extraction**    QA-ZRE introduced a similar idea for relation extraction (Levy et al., 2017). By associating natural language questions with relations, this dataset reduces relation extraction to question answering. This makes it possible to use question answering models in place of more traditional relation extraction models that often do not make use of the linguistic similarities amongst relations. This in turn makes it possible to do zero-shot relation extraction.

**Goal-Oriented Dialogue**    Dialogue state tracking requires a system to estimate a users goals and and requests given the dialogue context, and it plays a crucial role in goal-oriented dialogue systems. Most models use a structured approach (Mrkšić et al., 2016), with the most recent work making use of both global and local modules to learns representations of the user utterance and previous system actions (Zhong et al., 2018).

**Semantic Parsing**    Similarly, recent approaches to the semantic parsing WikiSQL dataset have made use of structured approaches that move from coarse sketches of the input to fine-grained structured outputs (Dong and Lapata, 2018), direclty employing a type system (Yu et al., 2018b), or making use of dependency graphs (Huang et al., 2018).

## D  Preprocessing and Training Details

All data is lowercased as is common for SQuAD, IWSLT, CNN/DM, and WikiSQL; casing is irrelevant for the evaluation of the other tasks. We use the RevTok tokenizer[4] to provide simple, yet completely reversible tokenization, which is crucial for detokenizing generated sequences for evaluation. The generative vocabulary in Eq. 11 contains the most frequent $50000$ words in the combined training sets for all tasks in decaNLP. SQuAD examples with context longer than $400$ tokens were excluded during training and CNN/DM examples had contexts truncated to $400$ tokens during training and evaluation. Only MNLI examples with a label other than '-' were included during training and evaluation as is standard. For WOZ, we train turn-by-turn to predict the change in belief state including user requests as an additional slot, but during evaluation we only consider the cumulative belief state as is standard. We do not perform any form of beam search or otherwise refine greedily sampled outputs for any tasks to avoid task-specific post-processing where possible.

The MQAN defined in Section 3 takes 300-dimensional GloVe embeddings trained on Common-Crawl (Pennington et al., 2014) as input. Words that do not have corresponding GloVe embeddings are assigned zero vectors instead. We concatenate 100-dimensional character n-gram embeddings (Hashimoto et al., 2016) to the GloVe embeddings. This corresponds to setting $d_{emb} = 400$ in Section 3. Internal model dimension $d = 200$, hidden dimension $f = 150$, and the number of heads in multi-head attention $p = 3$. MQAN uses 2 self-attention and multi-head decoder attention layers. We use a dropout of $0.2$ on inputs to LSTMs, layers following coattention, and decoder layers, before multiplying by $\tilde{Z}$ in Eq. 22, before adding $X$ in Eq. 25, and generally after any linear transformation. The models are trained using Adam with $(\beta_1, \beta_2, \epsilon) = (0.9, 0.98, 10^{-9})$ and a warmup schedule (Vaswani et al., 2017), which increases the learning rate linearly from 0 to $2.5 \times 10^{-3}$ over 800 iterations before decaying it as $\frac{1}{\sqrt{k}}$, where $k$ is the iteration count. Batches consist entirely of examples from one task and are dynamically constructed to fit as many examples as possible so that the sum of the number of tokens in the context and question and five times the number of tokens in the asnwer does not exceed $10000$.

---

[4]https://github.com/jekbradbury/revtok

# E    MULTITASK QUESTION ANSWERING NETWORK (MQAN) ENCODER

Recall from Section 3 that the encoder has three input sequences during training: a context $c$ with $l$ tokens, a question $q$ with $m$ tokens, and an answer $a$ with $n$ tokens. Each of these is represented by a matrix where the $i$th row of the matrix corresponds to a $d_{emb}$-dimensional embedding (such as word or character vectors) for the $i$th token in the sequence:

$$C \in \mathbb{R}^{l \times d_{emb}} \qquad Q \in \mathbb{R}^{m \times d_{emb}} \qquad A \in \mathbb{R}^{n \times d_{emb}} \tag{14}$$

**Independent Encoding.** A linear layer projects input matrices onto a common $d$-dimensional space.

$$CW_1 = C_{proj} \in \mathbb{R}^{l \times d} \qquad QW_1 = Q_{proj} \in \mathbb{R}^{m \times d} \tag{15}$$

These projected representations are fed into a shared, bidirectional Long Short-Term Memory Network (BiLSTM) (Hochreiter and Schmidhuber, 1997; Graves and Schmidhuber, 2005) [5]

$$\text{BiLSTM}_{ind}(C_{proj}) = C_{ind} \in \mathbb{R}^{l \times d} \qquad \text{BiLSTM}_{ind}(Q_{proj}) = Q_{ind} \in \mathbb{R}^{m \times d} \tag{16}$$

**Alignment.** We obtain coattended representations by first aligning encoded representations of each sequence. We add separate trained, dummy embeddings to $C_{ind}$ and $Q_{ind}$ (now $\in \mathbb{R}^{(l+1) \times d}$ and $\mathbb{R}^{(m+1) \times d}$) so that tokens are not forced to align with any token in the other sequence.

Let $\text{softmax}X$ denote a column-wise softmax that normalizes each column of the matrix $X$ to have entries that sum to $1$. We obtain alignments by normalizing dot-product similarity scores between representations of one sequence with those of the other:

$$\text{softmax}C_{ind}Q_{ind}^\top = S_{cq} \in \mathbb{R}^{(l+1) \times (m+1)} \qquad \text{softmax}Q_{ind}C_{ind}^\top = S_{qc} \in \mathbb{R}^{(m+1) \times (l+1)} \tag{17}$$

**Dual Coattention.** These alignments are used to compute weighted summations of the information from one sequence that is relevant to a single token in the other.

$$S_{cq}^\top C_{ind} = C_{sum} \in \mathbb{R}^{(m+1) \times d} \qquad S_{qc}^\top Q_{ind} = Q_{sum} \in \mathbb{R}^{(l+1) \times d} \tag{18}$$

The coattended representations use the same weights to transfer information gained from alignments back to the original sequences:

$$S_{qc}^\top C_{sum} = C_{coa} \in \mathbb{R}^{(l+1) \times d} \qquad S_{cq}^\top Q_{sum} = Q_{coa} \in \mathbb{R}^{(m+1) \times d} \tag{19}$$

The first column of the summation and coattentive representations correspond to the dummy embeddings. This information is not needed, so we drop that column of the matrices to get $C_{coa} \in \mathbb{R}^{l \times d}$ and $Q_{coa} \in \mathbb{R}^{m \times d}$.

**Compression.** In order to compress information from dual coattention back to the more manageable dimension $d$, we concatenate all four prior representations for each sequence along the last dimension and feed into separate BiLSTMs:

$$\text{BiLSTM}_{comC}([C_{proj}; C_{ind}; Q_{sum}; C_{coa}]) = C_{com} \in \mathbb{R}^{l \times d} \tag{20}$$

$$\text{BiLSTM}_{comQ}([Q_{proj}; Q_{ind}; C_{sum}; Q_{coa}]) = Q_{com} \in \mathbb{R}^{m \times d} \tag{21}$$

**Self-Attention.** Next, we use multi-head, scaled dot-product attention (Vaswani et al., 2017) to capture long distance dependencies within each sequence. Let

$$\text{Attention}(\tilde{X}, \tilde{Y}, \tilde{Z}) = \text{softmax}\left(\frac{\tilde{X}\tilde{Y}^\top}{\sqrt{d}}\right)\tilde{Z} \tag{22}$$

$$\text{MultiHead}(X, Y, Z) = [h_1; \cdots; h_p]W_o \qquad \text{where } h_j = \text{Attention}(XW_j^X, YW_j^Y, ZW_j^Z) \tag{23}$$

All linear transformations in Eq. equation 23 project to $d$ so that multi-head attention representations maintain dimensionality:

$$\text{MultiHead}_C(C_{com}, C_{com}, C_{com}) = C_{mha} \qquad \text{MultiHead}_Q(Q_{com}, Q_{com}, Q_{com}) = Q_{mha} \tag{24}$$

---

[5] For input $X \in \mathbb{R}^{T \times d_{in}}$, let $\overrightarrow{h_t} = \text{LSTM}\left(x_t^T, \overrightarrow{h_{t-1}}\right)$ and $\overleftarrow{h_t} = \text{LSTM}\left(x_t^T, \overleftarrow{h_{t+1}}\right)$. Representations are concatenated along the last dimension $h_t = [\overrightarrow{h_t}; \overleftarrow{h_t}]$ for each $t$ and stacked as rows of output $H \in \mathbb{R}^{T \times d_{out}}$.

We then use projected, residual feedforward networks (FFN) with ReLU activations (Nair and Hinton, 2010; Vaswani et al., 2017) and layer normalization (Ba et al., 2016) on the inputs and outputs. With parameters $U \in \mathbb{R}^{d \times f}$ and $V \in \mathbb{R}^{f \times d}$:

$$FFN(X) = \max(0, XU)V + X \tag{25}$$

$$FFN_C(C_{com} + C_{mha}) = C_{self} \in \mathbb{R}^{l \times d} \qquad FFN_Q(Q_{com} + Q_{mha}) = Q_{self} \in \mathbb{R}^{m \times d} \tag{26}$$

**Final Encoding.** Finally, we aggregate all of this information across time with two BiLSTMs:

$$\text{BiLSTM}_{finC}(C_{self}) = C_{fin} \in \mathbb{R}^{l \times d} \qquad \text{BiLSTM}_{finQ}(Q_{self}) = Q_{fin} \in \mathbb{R}^{m \times d} \tag{27}$$

These matrices are given to the decoder to generate the answer.

## F    CURRICULUM LEARNING

For multitask training, we experiment with various round-robin batch-level sampling strategies.

The first strategy we consider is fully joint. In this strategy, batches are sampled round-robin from all tasks in a fixed order from the start of training to the end. This strategy performed well on tasks that required fewer iterations to converge during single-task training (see Table 3), but the model struggles to reach single-task performance for several other tasks. In fact, we found a correlation between the performance gap between single and multitasking settings of any given task and number of iterations required for convergence for that task in the single-task setting.

With this in mind, we experimented with several anti-curriculum schedules Bengio et al. (2009). These training strategies all consist of two phases. In the first phase, only a subset of the tasks are trained jointly, and these are typically the ones that are more difficult. In the second phase, all tasks are trained according to the fully joint strategy.

We first experimented with isolating SQuAD in the first phase, and the switching to fully joint training over all tasks. Since we take a question answering approach to all tasks, we were motivated by the idea of pretraining on SQuAD before being exposed to other kinds of question answering. This would teach the model how to use the multi-context decoder to properly retrieve information from the context before needing to learn how to switch between tasks or generate words on its own. Additionally, pretraining on SQuAD had already been shown to improve performance for NLI (Min et al., 2017). Empirically, we found that this motivation is well-placed and that this strategy outperforms all others that we considered in terms of the decaScore. This strategy sacrificed performance on IWSLT but recovered the lost decaScore on other tasks, especially those which use pointers.

To explore if adding additional tasks to the initial curriculum would improve performance further, we experimented with adding IWSLT and CNN/DM to the first phase and in another experiment, adding IWSLT, CNN/DM and MNLI. These are tasks with a large number of training examples relative to the other tasks, and they contain the longest answer sequences. Further, they form a diverse set since they encourage the model to decode in different ways such as the vocabulary for IWSLT, context-pointer for SQuAD and CNN/DM, and question-pointer for MNLI. In our results, we however found no improvement by adding these tasks. In fact, in the case when we added SQuAD, IWSLT, CNN/DM and MNLI to the initial curriculum, we observed a marked degradation in performance of some other tasks including QA-SRL, WikiSQL and MWSC. This suggests that it is concordance between the question answering nature of the task and SQuAD that enabled improved outcomes and not necessarily the richness of the task.

Finally, as a check to our hypothesis, we also tried a curriculum schedule that used SST, QA-SRL, QA-ZRE, WOZ, WikiSQL and MWSC in the initial curriculum. This effectively takes the easiest tasks and trains on those first. This was indubitably an inferior strategy; not only does the model perform worse on tasks that were not in the initial curriculum, especially SQuAD and IWSLT, it also performs worse on the tasks that were. Finding that anti-curriculum learning benefited models in the decaNLP also validated intuitions outlined in (Caruana, 1997): tasks that are easily learned may not lead to development of internal representations that are useful to other tasks. Our results actually suggest a stronger claim: including easy tasks early on in training makes it more difficult to learn internal representations that are useful to other tasks.

We note in passing that the results above underscores the challenges and trade-offs in the multitasking setting. By ordering the tasks differently, it is possible to improve performance on some of the tasks but that improvement is not without a concomitant drop in performance for others. Indeed, a gap still exists between single-task performance and the results above. The question of how this gap can be bridged is a topic of continued research.

Table 3: Validation metrics for MQAN using various training strategies. The first is fully joint, which samples batches round-robin from all tasks. Others first use a curriculum or anti-curriculum schedule over a subset of tasks before switching to fully joint over all tasks. Curriculum first trains tasks that take relatively few iterations to converge when trained alone. This omits SQuAD, IWSLT, CNN/DM, and MNLI. The remaining strategies are anti-curriculum. They include in the first phase either SQuAD alone, SQuAD, IWSLT, and CNN/DM, or SQuAD, IWSLT, CNN/DM, and MNLI.

| | | | Anti-Curriculum | | |
|---|---|---|---|---|---|
| Dataset | Fully Joint | Curriculum | SQuAD | +IWSLT+CNN/DM | +MNLI |
| SQuAD | 70.8 | 43.4 | 74.3 | 74.5 | 74.6 |
| IWSLT | 16.1 | 4.3 | 13.7 | 18.7 | 19.0 |
| CNN/DM | 23.9 | 21.3 | 24.6 | 20.8 | 21.6 |
| MNLI | 70.5 | 58.9 | 69.2 | 69.6 | 72.7 |
| SST | 86.2 | 84.5 | 86.4 | 83.6 | 86.8 |
| QA-SRL | 75.8 | 70.6 | 77.6 | 77.5 | 75.1 |
| QA-ZRE | 28.0 | 24.6 | 34.7 | 30.1 | 37.7 |
| WOZ | 80.6 | 81.9 | 84.1 | 81.7 | 85.6 |
| WikiSQL | 62.0 | 68.6 | 58.7 | 54.8 | 42.6 |
| MWSC | 48.8 | 41.5 | 48.4 | 34.9 | 41.5 |
| decaScore | 562.7 | 499.6 | 571.7 | 546.2 | 557.2 |

# G    EXPANDED RESULTS

Table 4: Expanded Results for MQAN models with the addition of contextualized word vectors (CoVe, McCann et al. (2017)). The left-most column is the data used to train the MQAN model. When trained on decaNLP, the Fully Joint strategy was used.

| | SQuAD | IWSLT | CNN/DM | MNLI | SST | QA-SRL | QA-ZRE | WOZ | WikiSQL | MWSC | decaNLP |
|---|---|---|---|---|---|---|---|---|---|---|---|
| SQuAD | 77.2 | 0.24 | 6.46 | 0.00 | 0.00 | 52.2 | 29.3 | 0.00 | 0.00 | 22.0 | 187.4 |
| IWSLT | 2.52 | 27.7 | 7.32 | 0.00 | 0.00 | 8.81 | 0.00 | 0.00 | 0.00 | 0.00 | 46.4 |
| CNN/DM | 8.64 | 0.76 | 26.1 | 0.00 | 0.00 | 25.1 | 0.00 | 0.00 | 0.00 | 0.00 | 60.6 |
| MNLI | 1.36 | 0.00 | 0.29 | 74.2 | 31.0 | 0.06 | 0.22 | 0.00 | 0.00 | 11.9 | 119.0 |
| SST | 1.20 | 0.00 | 0.00 | 9.53 | 88.2 | 0.03 | 0.44 | 0.00 | 0.00 | 3.66 | 103.1 |
| QA-SRL | 17.7 | 0.00 | 7.76 | 0.00 | 0.00 | 79.2 | 4.67 | 0.00 | 0.00 | 18.3 | 127.6 |
| QA-ZRE | 7.66 | 0.00 | 0.45 | 0.00 | 0.00 | 3.64 | 27.1 | 0.00 | 0.00 | 1.22 | 40.1 |
| WOZ | 0.03 | 0.00 | 0.00 | 0.00 | 0.00 | 0.17 | 0.00 | 89.2 | 0.00 | 0.00 | 89.4 |
| WikiSQL | 4.52 | 0.38 | 14.0 | 0.33 | 0.00 | 13.85 | 0.44 | 0.00 | 72.9 | 1.22 | 107.6 |
| MWSC | 1.93 | 0.00 | 0.00 | 4.02 | 50.9 | 0.20 | 0.44 | 0.00 | 0.00 | 53.66 | 111.1 |
| decaNLP | 75.1 | 17.0 | 24.4 | 72.7 | 86.4 | 79.5 | 34.6 | 85.8 | 66.6 | 51.2 | 593.3 |

Table 5: Expanded Results for MQAN models. The left-most column is the data used to train the MQAN model. When trained on decaNLP, the SQuAD-first anti-curriculum strategy was used.

| | SQuAD | IWSLT | CNN/DM | MNLI | SST | QA-SRL | QA-ZRE | WOZ | WikiSQL | MWSC | decaNLP |
|---|---|---|---|---|---|---|---|---|---|---|---|
| SQuAD | 75.3 | 0.60 | 9.18 | 0.00 | 0.00 | 51.2 | 25.6 | 0.00 | 0.00 | 19.5 | 181.4 |
| IWSLT | 2.05 | 26.7 | 6.34 | 0.00 | 0.00 | 8.06 | 0.00 | 0.00 | 0.00 | 0.00 | 43.2 |
| CNN/DM | 8.67 | 0.71 | 25.5 | 0.00 | 0.00 | 25.2 | 0.00 | 0.00 | 0.00 | 0.00 | 60.1 |
| MNLI | 0.26 | 0.00 | 0.00 | 73.0 | 3.55 | 0.08 | 0.22 | 0.00 | 0.00 | 0.00 | 77.1 |
| SST | 0.78 | 0.00 | 0.03 | 11.8 | 88.5 | 0.05 | 0.00 | 0.00 | 0.00 | 3.66 | 104.8 |
| QA-SRL | 17.1 | 0.03 | 7.77 | 0.13 | 0.00 | 77.9 | 3.56 | 0.00 | 0.00 | 15.9 | 122.4 |
| QA-ZRE | 7.35 | 0.00 | 0.05 | 0.00 | 0.00 | 2.74 | 24.3 | 0.00 | 0.00 | 1.22 | 35.66 |
| WOZ | 0.18 | 0.00 | 0.04 | 0.00 | 0.00 | 0.48 | 0.00 | 88.0 | 0.00 | 0.00 | 88.7 |
| WikiSQL | 4.26 | 0.52 | 14.3 | 0.00 | 0.00 | 15.7 | 0.00 | 0.00 | 73.5 | 0.00 | 108.3 |
| MWSC | 1.99 | 0.00 | 0.00 | 28.1 | 50.9 | 0.08 | 0.44 | 0.00 | 0.00 | 48.8 | 130.3 |
| decaNLP | 74.4 | 18.6 | 24.3 | 71.5 | 87.4 | 78.4 | 37.6 | 84.8 | 64.8 | 48.8 | 590.6 |

## H    MODEL VISUALIZATION

Given that our networks are trained jointly, it is unclear whether the capacity of the network is implicitly provisioned for each task, or if there is sharing of neurons across tasks. To investigate this question, in Figure 5 we plot the activations of neurons at two encoder layers for both the context and question arms. For this experiment, we pick one representative example for 6 tasks and plot activations for all neurons at two layers: the output of the co-attention, and the final activations which are fed to the decoder. We use a trained MQAN model for this inference. As can be seen from the figure, there is a discernible pattern in the activations for the first layer of both arms but not for the deeper layer. The former is expected given that the co-attention tends to underscore weights that appear in both question and context. However, the lack of a discernible pattern in the deeper layer alludes to the notion that the capacity is not provisioned but shared.

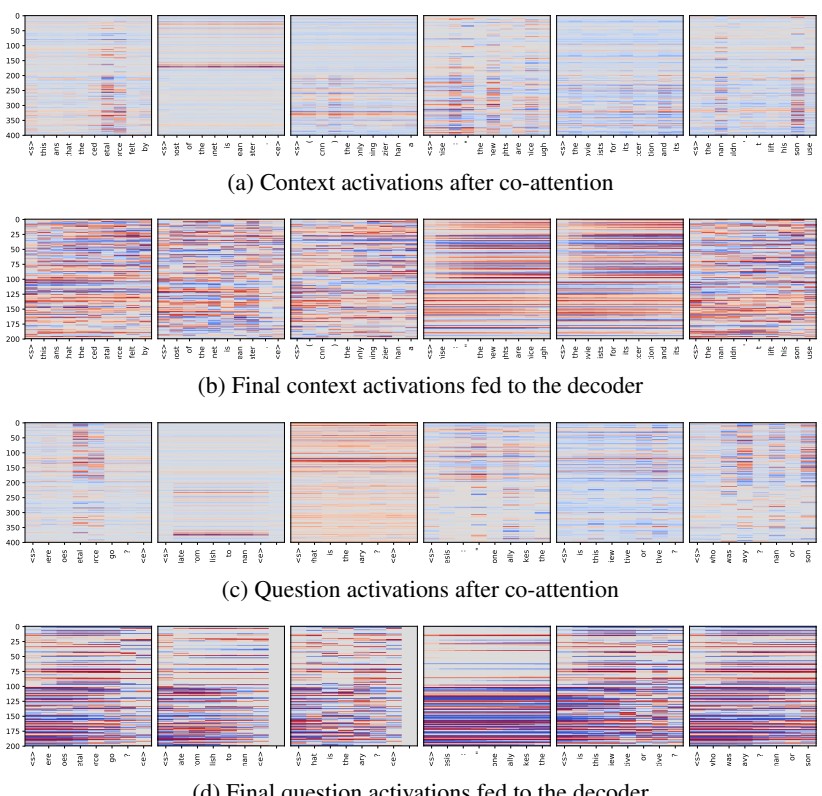

(a) Context activations after co-attention

(b) Final context activations fed to the decoder

(c) Question activations after co-attention

(d) Final question activations fed to the decoder

Figure 5: Visualization of encoder activations for a set of 6 (question, answer) pairs in the order: question answering, machine translation, summarization, natural language inference, and commonsense reasoning. x-axis for each block represents time, and y-axis denotes neurons in the layer.

In Figures 6 and 7, we plot the attention weights of the model over the context and question. The results are as one would expect, and are similar to those when training in single-task mode. It is evident from the figure that for most classification problems, there is a hard attention weight over the chosen (correct) answer.

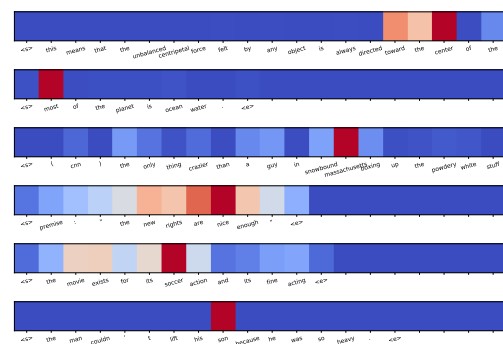

(a) Attention weights over the context at timestep 0

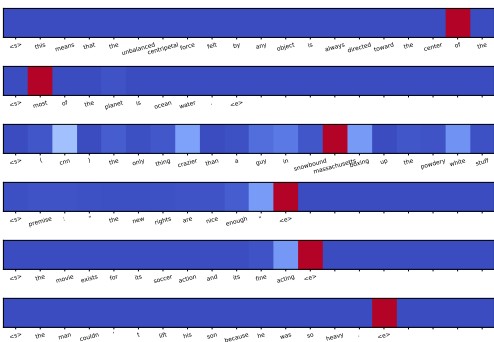

(b) Attention weights over the context at timestep 1

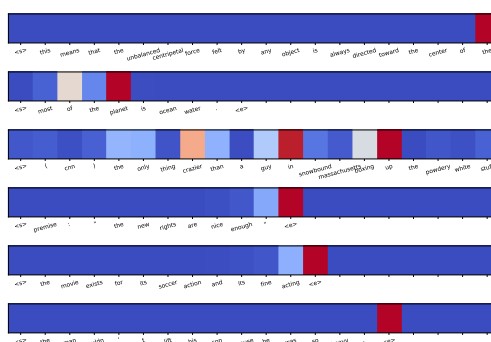

(c) Attention weights over the context at timestep 2

Figure 6: Visualization of attention weights over the context for a set of 6 (question, answer) pairs in the order: question answering, machine translation, summarization, natural language inference, and commonsense reasoning. x-axis for each block represents time.

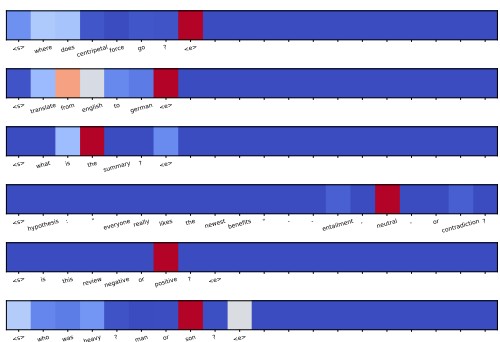

(a) Attention weights over the question at timestep 0

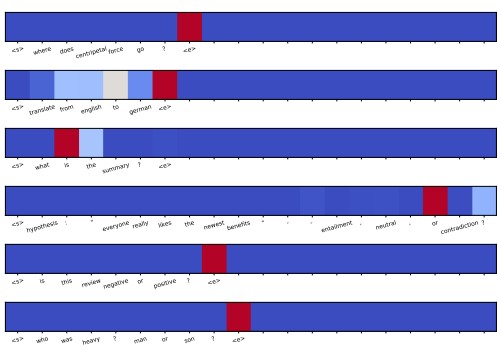

(b) Attention weights over the question at timestep 1

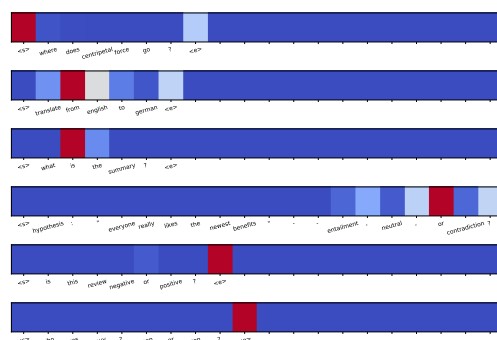

(c) Attention weights over the question at timestep 2

Figure 7: Visualization of attention weights over the question for a set of 6 (question, answer) pairs in the order: question answering, machine translation, summarization, natural language inference, and commonsense reasoning. x-axis for each block represents time.

