# OpenReview forum: "The Natural Language Decathlon: Multitask Learning as Question Answering"
_ICLR.cc/2019/Conference_

### Official Review · AnonReviewer2 · 2018-10-30
**Misguided and overcrowded**

**Rating:** 3
**Confidence:** 4

**Review:**

I appreciate the work that went into creating this paper, but I'm afraid I see little justification for accepting it.  I have three major complaints with this paper:

1. I think the framing of decaNLP presented in this paper does more harm than good, because it perpetuates a misguided view of question answering.

Question answering is not a unified phenomenon.  There is no such thing as "general question answering", not even for humans.  Consider "What is 2 + 3?", "What's the terminal velocity of a rain drop?", and "What is the meaning of life?"  All of these questions require very different systems to answer, and trying to pretend they are the same doesn't help anyone solve any problems.

Question answering is a _format_ for studying particular phenomena.  Sometimes it is useful to pose a task as QA, and sometimes it is not.  QA is not a useful format for studying problems when you only have a single question (like "what is the sentiment?" or "what is the translation?"), and there is no hope of transfer from a related task.  Posing translation or classification as QA serves no useful purpose and gives people the wrong impression about question answering as a format for studying problems.

We have plenty of work that studies multiple datasets at a time (including in the context of semi-supervised / transfer learning), without doing this misguided framing of all of them as QA (see, e.g., the ELMo and BERT papers, which evaluated on many separate tasks).  I don't see any compelling justification for setting things up this way.

2. One of the main claims of this paper is transfer from one task to another by posing them all as question answering.  There is nothing new in the transfer results that were presented here, however.  For QA-SRL / QA-ZRE, transfer from SQuAD / other QA tasks has already been shown by Luheng He (http://aclweb.org/anthology/N18-2089) and Omer Levy (that was the whole point of the QA-ZRE paper), so this is merely reproducing that result (without mentioning that they did it first).  For all other tasks, performance drops when you try to train all tasks together, sometimes significantly (as in translation, unsurprisingly).  For the Czech task, fine tuning a pre-trained model has already been shown to help.  Transfer from MNLI to SNLI is known already and not surprising - one of the main points of MNLI was domain transfer, so obviously this has been studied before.  The claims about transfer to new classification tasks are misleading, as you really have the _same_ classification task, you've just arbitrarily changed how you're encoding the class label.  It _might_ be the case that you still get transfer if you actually switch to a related classification task, but you haven't examined that case.

3. This paper tries to put three separate ideas into a single conference paper, and all three ideas suffer as a result, because there is not enough space to do any of them justice.  Giving 15 pages of appendix for an 8 page paper, where some of the main content of the paper is pushed to the appendix, is egregious.  Putting your work in the context of related work is not something that should be pushed into an appendix, and we should not encourage this behavior.

The three ideas here seem to me to be (1) decaNLP, (2) the model architecture of MQAN, (3) transfer results.  Any of these three could have been a single conference paper, had it been done well.  As it stands, decaNLP isn't described or motivated well enough, and there isn't any space left in the paper to address my severe criticisms of it in my first point.  Perhaps if you had dedicated the paper to decaNLP, you could have given arguments that the framing is worthwhile, and described the tasks and their setup as QA sufficiently (as it is, I don't see any description anywhere of how the context is constructed for WikiSQL; did I miss it somewhere?).  For MQAN, there's more than a page of the core new architecture that's pushed into the appendix.  And for the transfer results, there is very little comparison to other transfer methods (e.g., ELMo, CoVe), or any deep analysis of what's going on - as I mentioned above, basically all of the results presented are just confirming what has already been done elsewhere.

---

> ### Comment · AnonReviewer2 · 2018-11-26
> **Reply deleted**
>
> I received an email with a response; I'm assuming the authors posted the response and then deleted it, so that it only showed up to reviewers.  At the risk of escalating things further, I'm posting the reply here so I can respond to it.
>
> Response title: Red Flag
>
> Response comment: I'll respond to this review's points 2) and 3) eventually in a way that is visible to everyone without discussing this particular aspect of the review, but I have to say that the title and complaint 1) come off as condescending and, frankly, just plain mean.
>
> As a fellow ICLR reviewer, I can't imagine titling a review as "Misguided and Overcrowded". How about "Concerns with QA as a general framework; too much material in appendices"? That took me about a second to rephrase in a way that is more informative and avoids conveying an intention to humiliate and demean.
>
> Similarly, there are plenty of ways to politely raise concerns about multi-task learning and framing multi-task learning as question answering, but the reviewer chooses an alternative approach. Take for example this excerpt:
>
> "this paper does more harm than good, because it perpetuates a misguided view of question answering... Question answering is not a unified phenomenon.... There is no such thing as general question answering... All of these questions require very different systems to answer, and trying to pretend they are the same doesn't help anyone solve any problems."
>
> The above starts out by repeating the same baseless claim that I can find any number of people to disagree with. The way that paragraph ends makes it read as if the whole thing is really an ad hominem attack in disguise. In my opinion as a fellow reviewer, I do not think we should be entering into ideological arguments. Rather, the reviewer should be basing their claims in the empirical results of the paper and prior published literature. This reviewer is not doing that; they are just stating their opinion despite the fact that the idea of "general QA" has been used as an idea in this paper to get SOTA on WikiSQL and make significant progress on two crucial multi-task learning problems (see response to R3).
>
> I'm quite shocked that anyone that considers themselves part of a professional community would talk to someone else in that community so rudely. I'm not writing this so much as an author because the review eventually does raise some good concerns. I acknowledge that the paper has issues with the amount of information in the appendices. But -- as a reviewer I find myself asking, "Why did they have to have all the condescending meanness before getting to helpful, critical feedback? How does all that pontificating help the authors improve their research?" It is clear to me that it did not need to be there because such pontificating does not help. For this reason, I think this kind of review should be discouraged by ACs and Higher.

---

> > ### Comment · AnonReviewer2 · 2018-11-26
> > **Response**
> >
> > I apologize that my review came off to you as rude.  That was not my intent.  I knew that my review was quite negative, and I read it several times trying to make sure the criticisms were of the ideas in the paper, not the people who did the work.  I apparently did not do a good enough job of that, and I am sorry.  When I read it again, even having seen your response, I still have a hard time finding ad hominem attacks (and even you have to say that it's "in disguise").  I can imagine that when it's your work it feels more ad hominem than it was intended.
> >
> > I stand by my criticisms of the paper, however.  I strongly feel that this framing of translation and classification as QA harms QA research, and you have a very prominent, public voice advocating for it.  You say that my claim is "baseless" and you "can find any number of people to disagree with" it.  Citation please (or better yet, just give the arguments themselves instead of appealing to a nameless authoritative crowd).  I provided evidence in my second point - treating everything as QA makes performance on most tasks drop, except in cases where the task was designed to be QA and makes sense as QA.

---

> > > ### Author Response · Authors · 2018-11-26
> > > **Clarifications and thanks for your helpful feedback**
> > >
> > > Yes, I understand that your intent was probably not rudeness. I didn't think it was my place to publicly comment on your writing compared to say, R1, who makes nearly all the same criticisms and gives an equally low score without using terms that are condescending like 'misguided'. I did not post this publicly because I am both an author and a reviewer, and I understand that I am biased towards reading this review  as more negative than it should be read. That is why I posted this to ACs and Higher so that they could evaluate. For some reason, the system must have some unintuitive behavior (too me at least) that sends you an email for comments on your posts regardless of the chosen visibility. Not sure what happened since the original post is still visible to me and was not deleted. Now that you've posted it to Everyone, I might as well clarify.
> > >
> > > As a reviewer, my criticism of this review has nothing to do with QA or the paper itself. Title and 1) seem to be written too combatively (perhaps to use this platform to balance out "very prominent, public voice[s] advocating for it"?). I don't think this is the place for that. On my view, authors submit for review to get valuable criticism. The reviewer's ultimate goal should be to tell authors how to improve their research; it should not be to combat the research agenda. The paper has problems, especially in total content and organization. As mentioned in the post to ACs and Higher, you raise important criticisms in points 2) and 3). But, I think Title and 1) deviated from what I see as a reviewer's goal too much. I just think you could have done without 1), and you should also avoid using words like 'misguided' unless you intend to up the rudeness factor by a few notches. We might just have to agree to disagree here about tone and word choice, maybe even about the goal of reviewing.
> > >
> > > Now switching back into author mode.
> > >
> > > 2) You're right that the transfer learning experiments for any one task are not new results. What we find interesting here is that the multi-task model retains transferability to all of the tasks it has been trained on. In this sense, these experiments verify that the representations of the multi-task model are somehow compressing the transferable utility of ten single-task models into a single model (10x smaller).
> > >
> > > Regarding your point about the gap between single- and multi-task performance, I'll point you to our response to R3 so that you don't have to do redundant reading.
> > >
> > > Regarding switching classification labels. Yes, this is a rough approximation for something we were trying to study -- whether the model could adapt to new, but related kinds of questions and adapt its output space. Certainly this experimental design has some problems, but we do think it demonstrates the more general capacity of the model to switch output spaces based on the question because the model must realize that even though the context is the same, it must use different output labels based on different questions.
> > >
> > > 3) No objections here. Organizing all this information into a reasonable order is tough, and clearly one big take-away from this reviewing process is to break things down into more conference-sized chunks rather than cram everything into appendices. Definitely don't put related works in an appendix -- it is disrespectful even if the intentions were good (more space to expand on it all).
> > >
> > > Final paragraph) A lot of additional valuable feedback here. This gives a good sense of how we might restructure and support claims with new experiments. Very much appreciated.
> > >
> > > Overall, thanks for the discussion. Even though I disagreed with your reviewing style for 1), I think you make really good points in the remainder of your review. Thanks for offering so much of your time.

---

> > > > ### Comment · AnonReviewer2 · 2018-11-27
> > > > **A better attempt at point 1**
> > > >
> > > > It seems my idiolect has a different connotation for "misguided" than yours, and I apologize for using a term that was offensive.  What I meant was essentially, "fundamentally the wrong way of thinking about the problem."  If I'm not supposed to comment on the framing of a research problem in a review, I'm not sure what the point of the review is.  You called my paragraphs in point 1 "pontificating" - I read them as arguments explaining _why_ I think this is the wrong way of thinking about the problem.  I have seen no counter-arguments from you, either in the paper or in your response to my review.
> > > >
> > > > So, some constructive criticism: provide me some arguments for why we should be thinking about "question answering" as a general phenomenon, or show empirically that we can get some benefit from thinking about things this way.  I see no empirical results that demonstrate that this is worthwhile, in fact I see quite the opposite.  While ELMo and BERT improve performance through multi-task learning, treating everything as QA and training them jointly hurts performance in almost all cases.
> > > >
> > > > You've mentioned SOTA on WikiSQL, but recall that those results were from _single task_ performance of MQAN and have nothing to do with transfer.  Performance unsurprisingly drops, quite a lot, when you try to jointly train WikiSQL with other "QA" tasks.
> > > >
> > > > If you're able to show that some gains can be had for translation or classification by thinking of them as QA (more than you can get by doing the same kind of label replacement but without QA), then I will be quite happy to give your paper a positive review.  Until then, this really feels like it's going down the wrong path and will give people the wrong impression about QA research.  I have had conversations with senior researchers who do not take QA research seriously because of papers saying that "everything is QA" - this is not theoretical harm that I am talking about.

---

> > > > > ### Comment · Area_Chair1 · 2018-11-30
> > > > >
> > > > > Dear Reviewer 2 and Authors
> > > > >
> > > > > The aim of the peer review process is to ensure that the work presented at conferences is of a sufficient scientific standard. To this end, while not necessarily so, it can end up being an adversarial process: results must be examined, comparisons must be called for, assumptions must be questioned, and so on. We must not let these moments where constructive criticism, and even rejection, is called for poison the well of communication, community, and collaboration which permits our field to grow. To this end, it is extremely important that the authors of negative reviews be especially mindful of their language and of how criticism is framed.
> > > > >
> > > > > Upon examining Reviewer 2's initial comments, I agree with the authors that—while not explicitly insulting—the tone is unpleasant. Reviewer 2 perhaps did not intend this, and has apologized for any offense caused. The content of the review is detailed and objective enough that I am not worried about the authors being treated unfairly, when it comes to the assessment of the paper. I encourage the reviewer to consider one last time their score in view of the discussion that has been had, and other reviews, and consider whether they wish to keep it (if so, why) or adjust it. I also encourage the reviewer to consider, in future reviews, how their well-meaning and expert counsel might be perceived by authors—who may perhaps be students or otherwise fairly new to our field—if improperly presented.
> > > > >
> > > > > AC

---

> ### Public Comment · (anonymous) · 2024-11-26
> **revisiting this review in 2024**
>
> I came across this review today and found this comment to be quite interesting.
>
> "Question answering is not a unified phenomenon.  **There is no such thing as "general question answering", not even for humans.**  Consider "What is 2 + 3?", "What's the terminal velocity of a rain drop?", and "What is the meaning of life?"  **All of these questions require very different systems to answer**, and trying to pretend they are the same doesn't help anyone solve any problems." -- ICLR 2019 Conference Paper1522 AnonReviewer2

---

### Official Review · AnonReviewer1 · 2018-11-02
**New framework has a lot of potential, but the experiments, motivations, and related work are missing details**

**Rating:** 5
**Confidence:** 3

**Review:**

Update: I've updated my score based on the clarifications from the authors to some of my questions/concerns about the experimental set-up and multi-task/single-task differences.

Original Review:
This paper provides a new framework for multitask learning in nlp by taking advantage of the similarities in 10 common NLP tasks. The modeling is building on pre-existing qa models but has some original aspects that were augmented to accommodate the various tasks.  The decaNLP framework could be a useful benchmark for other nlp researchers.

Experiments indicate that the multi-task set-up does worse on average than the single-task set-up.  I wish there was more analysis on why multi-task setups are helpful in some tasks and not others.  With a bit more fine-grained analysis, the experiments and framework in this paper could be very beneficial towards other researchers who want to experiment with multi-task learning or who want to use the decaNLP framework as a benchmark.

I also found the adaptation to new tasks and zero-shot experiments very interesting but the set-up was not described very concretely:
  -in the transfer learning section, I hope the writers will elaborate on whether the performance gain is coming from the model being pretrained on a multi-task objective or if there would still be performance gain by pretraining a model on only one of those tasks.  For example, would a model pre-trained solely on IWSLT see the same performance gain when transferred to English->Czech as in Figure 4? Or is it actually the multi-task training that is causing the improvement in transfer learning?
  -Can you please add more detail about the setup for replacing +/- with happy/angry or supportive/unsupportive? What were the (empirical) results of that experiment?

I think the paper doesn’t quite stand on its own without the appendix, which is a major weakness in terms of clarity.  The related work, for example, should really be included in the main body of the paper.  I also recommend that more of the original insights (such as the experimentation with curriculum learning) should be included in the body of the text to count towards original contributions.

As a suggestion, the authors may be able to condense the discussion of the 10 tasks in order to make more room in the main text for a related work section plus more of their motivations and experimental results.  If necessary, the main paper *can* exceed 8 pages and still fit ICLR guidelines.

Very minor detail: I noticed some inconsistency in the bibliography regarding full names vs. first initials only.

---

> ### Author Response · Authors · 2018-11-26
> **Response to R1: Thanks for your review, and some clarificaitons**
>
> Regarding your point about the gap between single- and multi-task performance, I'll point you to our response to R3 so that you don't have redundant reading.
>
> Regarding the transfer learning experiments. The performance gain does not come from the multi-task objective, as single-task models would exhibit similar behavior for their respective task. What we find interesting here is that the multi-task model retains transferability to all of the tasks it has been trained on. In this sense, these experiments verify that the representations of the multi-task model are somehow compressing the transferable utility of ten single task models into a single model (10x smaller!).
>
> For the label replacement on the SST dataset, the empirical results show a minor degradation in performance (~1%, so ~86 vs ~87 according to Table 2 and subsection 4.3). This was a naive replacement mapping all answers that were 'positive' to 'happy' and all answers that were 'negative' to 'angry'. This shows how the model is learning to capitalize on the common output space (all of English in GloVe) to adapt to new labels without any additional training. This is advantageous over models that do not actually generate answer sequences because it allows them to be more robust in intuitive ways.
>
> You're certainly right that the appendix carries a lot of useful information and some of the details about contributions. We had moved the related works to the appendix because that was the only way we found we could sufficiently do justice to the long line of literature in multi-task learning as well as all of the literature for each task, but it does seem we will need to include at least a part of our full related works in the main body. There is quite a bit of material overall, and we thank you for your suggestions about where to cut/condense and how to prioritize information.
>
> Thank you again for your questions and your feedback about organization.

---

> > ### Comment · AnonReviewer1 · 2018-11-30
> > **Response for authors**
> >
> > Thank you for clarifying!  I agree with several of the points you make above, and I appreciate your argument about the potential of the multi-task set-ups for transferability and compression.  I hope that you are able to revise future iterations of the paper to reflect some of the strong points you've made in the comments section here.

---

> ### Comment · Area_Chair1 · 2018-11-30
> **Related work sections**
>
> I am uncomfortable with this assessment. The reviewer is right that the related work section should not be in the appendix. The reviewer is also correct that the paper should stand on its own without the supplementary material. The role of the paper is to advertise and motivate the work, describing the key experiments, and the supplementary exists to provide enough detail for, say, reproduction or further analysis. Authors should not take advantage of supplementary materials to compensate for a poorly written or organized paper, or to bypass page limits while preserving large swathes of material in overall their submission.
>
> From looking over this paper, and without prejudice to whatever faults it may or may not otherwise have, it is clear that while the authors made a mistake in moving the contents of Appendices B and C out of the main paper, it was clearly not done in bad faith. The paper is under 8 pages, and the content of these appendices could clearly be moved into the main paper with minor and workable changes while remaining under 10 pages. It seems unfair to me to strongly argue the paper is worthy of rejection on these grounds.
>
> Please, could Reviewer 1 explain in further detail why they are recommending clear rejection: other than the relevant work sections, are there any sections currently in the appendix for which the paper suffers by not having them in the main body? Are there any other strong grounds for rejection? I must admit it is not clear to me from reading the review, in its current form.
>
> You are welcome to discuss these issues with the authors and other reviewers, as there is about a week left before I must provide preliminary decisions.
>
> AC

---

> > ### Comment · AnonReviewer1 · 2018-11-30
> > **response to AC**
> >
> > Thanks for commenting!  I'm sorry if my review was unclear.
> >
> > I agree that it seems that most of what they moved to the appendix was not done in bad faith.  I'm sorry if my review suggested that was the case.  However, I did think that there were some interesting details (aside from the related work) mentioned briefly in the appendix that I would have appreciated more analysis on, such as the experiments on curriculum learning strategies or the comparison of which tasks were more similar to each other (and therefore more beneficial for multi-task learning).  But, again, I don't think that was done in bad faith.  I really just wanted to provide feedback for the authors on that point.  I was not recommending rejection on those grounds.
> >
> > Rather, I had some difficulty determining the original contributions of this paper.  One problem for me was that some of the experiments were unclear (which I asked the authors about in my review), which made it difficult to understand what we can conclude from them.  This was particularly the case in the transfer-learning experiment which seemed to suggest that the benefits in transfer learning were coming from the multi-task set-up directly, without showing a single-task transfer-learning baseline (which the authors responded to in their review).
> >
> > Another question I had was about what the advantages of multi-task/single-task set-ups were.  In the paper's tables, it is clear that multi-task set-ups are outperformed by the single-task set-up in nearly all tasks (as well as overall by a nontrivial margin).  This goes against the main point of the paper (which seems to be that multi-task setups are beneficial), but it isn't discussed much in the running text.  I was hoping the authors would clarify a bit more about why we should use multi-task set-ups if single-task set-ups typically outperform them.  Because of the discrepancy in performance, I would have appreciated a more detailed discussion/analysis of the advantages of multi-task learning (this is brought up by Reviewer 3 as well).
> >
> > I appreciate the response/clarifications of the authors to many of my comments and questions.  I'm not sure that I could recommend a strong acceptance, but I would probably be inclined to raise my initial rating slightly based on their clarifications.

---

> > > ### Comment · Area_Chair1 · 2018-11-30
> > > **response**
> > >
> > > Thank you for giving further details of your concerns.
> > >
> > > I do not wish for you to think that my comment was compelling you to change your score, although you are welcome to do so if you think it right.
> > >
> > > Ultimately, you should give a score which you think reflects the strength and suitability of the paper for the conference. The only thing that matters to me (and, I suspect, to the authors) is that if you are going to recommend rejection, it be for clear reasons and with sufficient detail to permit the authors to properly revise their paper for further resubmission.
> > >
> > > AC

---

> > > > ### Comment · AnonReviewer1 · 2018-11-30
> > > > **response to AC**
> > > >
> > > > Thank you for replying.  I understand the point you are making.  I have updated my scoring because, after re-reading the author responses, I think my rating needed to be updated to reflect their clarifications.  Thanks, again, for pointing this out.

---

### Official Review · AnonReviewer3 · 2018-11-03
**A good example to treat different NLP problems as Q&A and trained together. Results for some problems are worse than their state-of-the-art.**

**Rating:** 5
**Confidence:** 4

**Review:**

The paper formulates several different NLP problems as Q&A problem and proposed a general deep learning architecture. All these tasks are trained together.

If the goal is to achieve general AI, the paper gives a good starting point. One technical novelty is the deep learning architecture for this general Q&A problem including the multi-pointer-generator. The paper presents an example of how to do a multi-task learning for 10 different tasks. It raises a very challenging problem or in some way release a new dataset.

If our goal is to optimize a single task, the usefulness of the method proposed by the paper is questionable.
As we know, multi-task learning works well if some important knowledge shared by different tasks can be learned and leveraged. From table 2, we see for many problems, the results of the single task training are better than the multi-task training, meaning that other tasks can't really help at least under this framework. This makes me doubt if this multi-task learning is useful if our goal is to optimize the performance of a single task. This general model also sacrifices some important prior knowledge of an individual task. For example, for the Squad, the prior that the answer is a continuous span. Ideally, the prior knowledge should be leveraged.

---

> ### Author Response · Authors · 2018-11-26
> **Respone to R3: Thank you for your review; more on single- vs. multi-task performance**
>
> First of all, thank you for your review. You touch upon a crucial point that does require clarification: the gap between the single- and multi-task performance.
>
> As you mentioned, the multi-task learning literature has taught us at least one thing: related tasks tend to help each other, and unrelated tasks tend to interfere with each other. The latter is an interesting phenomenon, and it is what we see as the primary multi-task learning problem of concern in this paper, and we are proposing decaNLP as a benchmark for measuring progress on this problem.
>
> There are two ways in which unrelated tasks tend to interfere. The first is during the modeling phase where some tasks prevent us from using priors (like span prediction for QA or a German-only output vocabulary) that would be useful for some tasks. The second is during the training phase where some tasks tend to interfere with representation learning.
>
> These two kinds of interference lead to two kinds of gaps that we measure with this benchmark. The first is the gap between the current best decaNLP model (in the single- and multi-task settings) and a combination of state-of-the-art models for each task. The second is the gap between the best decaNLP model in the multi-task setting and a combination of ten of those best decaNLP models each trained for a single task.
>
> The concrete contributions of this paper are 1) the preparation of benchmark along with reasonable sequence-to-sequence baselines, 2) progress on the first kind of gap by switching from seq2seq to multi-sequence-to-sequence with MQAN (by transforming problems into QA triplets), and 3) progress on the second kind of gap by demonstrating the superiority of anti-curriculum learning (or pre-training on harder tasks) over the baseline fully joint training. 3) actually ties multi-task learning back to transfer learning as an effective means of representation learning.
>
> But yes, we have not yet entirely closed these gaps; as you mentioned, that is a key part of the challenge to the community. We have chosen to introduce this challenge now because we believe solutions to this problems are within reach in the near future if the community focuses on them.
>
> And yes, though this approach will likely be successful whenever tasks are related (just based on what we know from the rest of the multi-task learning literature), it is sometimes not yet the best way to optimize for single-task performance. Keep in mind though that it did lead to new state-of-the-art results on WikiSQL despite no direct modeling or tuning for that task.
>
> Thanks again for your time.

---

> > ### Comment · AnonReviewer3 · 2018-11-28
> > **comments to authors**
> >
> > I agreed on the point that the paper raises an interesting challenge and a potentially interesting research direction. I also agree that not any set of tasks can be combined together for the multi-task learning. More analysis and study should be done to decide which tasks can benefit each other. I am interested in seeing that authors give more study in this direction and/or narrow the gaps (as mentioned in the response) in the future work.

---

> > > ### Author Response · Authors · 2018-11-28
> > > **Thanks for the feedback!**
> > >
> > > We’ll keep working on the gaps and make sure to provide additional analysis of task relatedness in future work.

---

### Public Comment · ~quan_vuong1 · 2018-10-09
**Clarifications**

Thanks for the paper and the collections of datasets!

I'm using decaNLP in my research and would like to ask a clarification question. Section 3 mentions "We gives it access to v additional vocabulary tokens". What are the v additional tokens and how were they chosen?

---

> ### Author Response · Authors · 2018-11-26
> **50k most common words across all tasks in decaNLP**
>
> Exciting! Glad you're finding decaNLP to be a good resource for further research!
>
> In our experiments, the generative vocabulary in Eq. 11 contains the most frequent 50000 words in the
> combined training sets for all tasks in decaNLP. A lot of these training details are way down in Appendix D on Preprocessing and Training Details. They aren't necessarily optimal if you have a bigger memory budget than we did or have a more clever motivation for how these kinds of decsision should be based on individual tasks.

---

### Public Comment · (anonymous) · 2018-10-09
**Please respect prior work**

The related work section should not be buried in Appendix B on page 17.

From the text of the main paper, a reader would have no indication that multi-task NLP has been around for 10+ years, and that the main novelty here is the particular selection of tasks and aggregating performance across those tasks as a benchmark. The authors should be more clear and honest about what their contribution is.

As an example, I'll point to [1], a well known paper (2.7k cites) from ICML 2008, titled "A Unified Architecture for Natural Language Processing: Deep Neural Networks with Multitask Learning".

[1] https://ronan.collobert.com/pub/matos/2008_nlp_icml.pdf

---

> ### Author Response · Authors · 2018-11-26
> **Related work**
>
> Thanks for taking the time to suggest how we could prioritize all of this information more effectively.
>
> You're right that even though we cite the work you mention (Collobert and Weston 2008) along with the follow up (Collobert et al. 2011) in our original submission, we only do so in the related works, which are currently placed in the appendices.
>
> I assure you that we meant no disrespect to these related works by placing them in an appendix. Our thinking at the time was that we could only do justice to the significant literature in both multi-task learning and single-task learning for all these tasks by moving such discussions to sections that had no page limit.
>
> We have a lot of information in the appendices that we view as just as important as the information in the main body. We just didn't have the same interpretation of appendices (as lesser material) going into this submission. We simply ordered on what we thought would need to be read first in order to understand the benchmark and the progress so far. For example, the details on anti-curriculum pre-training are actually quite important to us as a contribution, but they didn't seem as essential as understanding the nature of all the tasks. Our motivations for the tasks, the related works, and the task-specific related works are all important. The fact that they are in appendices is only because of the total amount of information in the submission.
>
> That being said, feedback from multiple sources has indicated that at least some of these related works need to be in the main body, and we will reorganize the paper accordingly.

---

### Meta-Review · Area_Chair1 · 2018-12-13
**Some great contributions, but more work needed on cross-task transfer**

**Confidence:** 5
**Recommendation:** Reject

**Metareview:**

This paper presents a new multi-task training and evaluation set up called the Natural Language Decathlon, and evaluates models on it. While this AC is sympathetic to any work which introduces new datasets and evaluation tasks, the reviewers agreed amongst themselves that the paper is not quite ready for publication. The main concern is that multi-task learning should show benefits of transferring representations or other model components between tasks, demonstrating better generalisation and less task-specific overfitting, but that the results in the paper do not properly show this effect. A more thorough study of which tasks "interact constructively" and what model changes can properly exploit this needs to be done. With this further work, the AC has no doubt that this dataset and task suite, and associated models, will be very valuable to the NLP community.

I should note that there were some issues during the review period which lead to AC-confidential communication between AC and authors, and AC and reviewers, to be leaked to the reviewers. It was due to an OpenReview bug, and no party is at fault. Through private discussion with the interested parties, we were able to resolve this matter, and through careful examination of the discussion, I am satisfied that the reviews and final recommendations of the reviewers were properly argued for and presented in good faith.